# Cadmium (II)-Induced Oxidative Stress Results in Replication Stress and Epigenetic Modifications in Root Meristem Cell Nuclei of *Vicia faba*

**DOI:** 10.3390/cells10030640

**Published:** 2021-03-13

**Authors:** Aneta Żabka, Konrad Winnicki, Justyna Teresa Polit, Mateusz Wróblewski, Janusz Maszewski

**Affiliations:** Department of Cytophysiology, Faculty of Biology and Environmental Protection, University of Lodz, 90-236 Lodz, Poland; konrad.winnicki@biol.uni.lodz.pl (K.W.); justyna.polit@biol.uni.lodz.pl (J.T.P.); mateusz.wroblewski@edu.uni.lodz.pl (M.W.); janusz.maszewski@biol.uni.lodz.pl (J.M.)

**Keywords:** cadmium, DNA damage, chromatin structure, epigenetic modifications, gamma-H2AX, H3S10Ph, H4K5Ac, oxidative stress, replication stress, *Vicia faba*

## Abstract

Among heavy metals, cadmium is considered one of the most toxic and dangerous environmental factors, contributing to stress by disturbing the delicate balance between production and scavenging of reactive oxygen species (ROS). To explore possible relationships and linkages between Cd(II)-induced oxidative stress and the consequent damage at the genomic level (followed by DNA replication stress), root apical meristem (RAM) cells in broad bean (*V. faba*) seedlings exposed to CdCl_2_ treatment and to post-cadmium recovery water incubations were tested with respect to H_2_O_2_ production, DNA double-strand breaks (γ-phosphorylation of H2AX histones), chromatin morphology, histone H3S10 phosphorylation on serine (a marker of chromatin condensation), mitotic activity, and EdU staining (to quantify cells typical of different stages of nuclear DNA replication). In order to evaluate Cd(II)-mediated epigenetic changes involved in transcription and in the assembly of nucleosomes during the S-phase of the cell cycle, the acetylation of histone H3 on lysine 5 (H3K56Ac) was investigated by immunofluorescence. Cellular responses to cadmium (II) toxicity seem to be composed of a series of interlinked biochemical reactions, which, via generation of ROS and DNA damage-induced replication stress, ultimately activate signal factors engaged in cell cycle control pathways, DNA repair systems, and epigenetic adaptations.

## 1. Introduction

Morphogenetic processes in plants are determined by cell divisions in the primary shoot and root apical meristems (SAMs and RAMs, respectively) and in the secondary (lateral) meristems, the cambia. All these regions, comprising cell populations characterized by a high mitotic activity, are exposed to and endangered by a variety of stress conditions, including those resulting from environmental contamination with heavy metals. Although some of them are essential micronutrients for plant growth and development (Cu, Zn, Fe, Mn, Mo, Ni, and Co), other non-essential elements, such as Cd and Pb (at elevated concentrations), may negatively affect the morphology, physiology, and biochemistry of plant tissues [1,2,3]. The toxicity of heavy metals includes root and stem growth inhibition, reduction in biomass accumulation, decrease in membrane stability and photosynthesis rate, compromised pigment production, and enhanced generation of reactive oxygen species (ROS) [4,5]. Once inside the cell, heavy metals activate different signaling pathways, which detect stress factors, transmit molecular signals from the cell exterior to cell interior, and initiate appropriate mechanisms to reduce the negative effects of stressors [6].

Unperturbed DNA biosynthesis determines the integrity of chromatin, and hence any disturbance in its structure may become a source of replication stress (RS) [7]. Chromatin reorganization caused by RS-induced DNA damage or an unbalanced pool of nucleotides may result in disadvantageous changes in gene expression and a consequent decrease in cellular metabolism. Stress factors that trigger such processes include cadmium, which is one of the most dangerous heavy metals (even if present in very low concentrations), responsible for numerous deleterious effects observed in plants at the genetic, biochemical, and physiological levels [8,9]. Since cadmium is chemically stable and does not undergo degradation [10], once released into the environment, it continues to circulate. This feature is combined with the high toxicity of cadmium and its ability to accumulate in plant organisms.

Cadmium enters plants as a divalent cation (Cd^2+^), and after being easily absorbed by the roots through low-specificity Ca^2+^ transporters or channels, it is transported to other organs, such as shoots [11]. Due to direct access to this metal in the soil, reduction or inhibition of root growth is one of the earliest symptoms of cadmium toxicity. Cd^2+^ ions affect photosynthetic activity, respiration, nitrogen metabolism, mineral uptake, and seed germination [12,13,14,15]. Moreover, the presence of cadmium changes the permeability of cell membranes, leading to water imbalance [2,16]. After entering the cell, Cd^2+^ disturbs protein function by directly attacking the thiol, histidyl, and carboxyl groups, and contributes to the excessive production of reactive oxygen species (ROS) [17]. At lower concentrations, Cd^2+^ enhances DNA replication, differentiation, and cell proliferation, while at higher doses it inhibits DNA synthesis and cell division, and induces changes related to cell death [16]. The effects of cadmium on cell cycle processes in roots and leaves are often accompanied by changes in the expression of genes encoding cell cycle-specific proteins, such as cyclin-dependent kinases (CDKs) and cyclins, thus leading to disturbances in the control over both G1-to-S-phase and G2-to-M-phase transitions [18,19].

By using primary root meristems of *Vicia faba* as an experimental model, our present study was aimed at finding possible relationships and linkages between cadmium (the primary cause of observable effects) and two types of consequent cadmium-induced secondary stress conditions, i.e., oxidative stress, which by reactive oxygen species (ROS) may play an important part in producing disorder at the genomic DNA level, leading to replication stress. To achieve this, RAM cells in *V. faba* seedlings exposed to 24 h CdCl_2_ treatment (at a concentration of 150 µM) and post-cadmium recovery incubations with water (12 and 24 h) were tested with respect to mitotic activity, chromatin morphology, ROS production (using 3,3-diaminobenzidine (DAB) polymerization-linked staining), γ-phosphorylation of H2AX histones on serine 139 (as a marker of DNA double-strand breaks, DSBs), phosphorylation of H3 histones on serine 10 (a marker of chromatin condensation), and EdU staining (to quantify cells typical of different stages of nuclear DNA replication). In order to evaluate cadmium-induced epigenetic changes involved in the assembly of nucleosomes during DNA replication and repair processes, also localized in the promoter sequences of active genes (to enhance transcriptional processes), acetylation of histone H4 on Lys 5 (H4K5Ac) was investigated by immunofluorescence.

## 2. Materials and Methods

### 2.1. Plant Material

Sterile seeds of field bean (*Vicia faba* subsp. minor L.) were sown on moist blotting paper and germinated in the dark at 20 ± 21 °C. Four-day-old seedlings with roots (2.5 ± 0.5 cm in length) were placed in Petri dishes containing distilled water (control samples) and 150 µM CdCl_2_ solution, and cultivated for 24 h in the dark. The concentration of CdCl_2_ was chosen based on results obtained in tests selected from available literature data (e.g., [20,21]) and a series of preliminary tests (not shown). Two portions of cadmium-treated seedlings were post-incubated with water (post-cadmium recovery): (1) for 12 h (Cd-rec-12) and (2) for 24 h (Cd-rec-24).

### 2.2. Detection of H_2_O_2_ Using DAB

Hydrogen peroxide (H_2_O_2_) was detected using 3,3-diaminobenzidine tetrachloride (DAB-HCl; Sigma-Aldrich, Poznan, Poland) according to procedures described earlier [22,23]. Seedlings (4 experimental series: control, CdCl_2_-treated, and post-cadmium water-incubated plants) were incubated in 1 mg mL^−1^ DAB-HCl in TRIS buffer (10 mM Tris, 10 mM EDTA-2Na, 100 mM NaCl; pH 7.5) dissolved in water (control and water-incubated recovery) or with the addition of 150 µM CdCl_2_. The histochemical reaction was stopped with distilled water. After staining, roots were fixed for 20 min in 4% paraformaldehyde dissolved in phosphate-buffered saline (PBS), washed three times with PBS, and macerated using citric acid-buffered 2.5% pectinase solution (pH 5.0; 37 °C for 15 min). Root apical meristems were squashed onto microscope slides and then stored in a mixture of glycerol and PBS (9:1; *v*/*v*). The presence of H_2_O_2_ was visualized as brown aggregates of polymerized DAB; mean intensity of DAB staining was measured by using computer-aided cytometry.

### 2.3. Feulgen DNA Staining

Roots excised from 5-day-old *V. faba* seedlings were fixed in ice-cold Carnoy’s solution (absolute ethanol and glacial acetic acid; 3:1, *v*/*v*) for 1 h, washed several times with ethanol, then rehydrated and hydrolyzed in 4 M HCl (2 h). The staining procedure with pararosaniline (Schiff’s reagent) was performed according to the standard method (e.g., [24]). After rinsing in SO_2_–water (3 times) and distilled water, RAMs (1.5 mm long) were cut off, placed in 45% acetic acid, and squashed onto Super-Frost microscope slides. Following freezing using dry ice, coverslips were removed, and the dehydrated slides were mounted in Canada balsam.

To evaluate mean values of mitotic indices and to quantify the occurrence of cells with aberrant chromosome structures, 10,000 Feulgen DNA-stained cells (taken from 10 root meristems) were analyzed for each individual mean value.

### 2.4. EdU Labeling

The control and all cadmium-treated seedlings of *V. faba* were incubated with 10 μM 5-ethynyl-2′-deoxyuridine (EdU; Thermo Fisher Scientific, Warsaw, Poland) for 20 min, in the dark. The incubation medium prepared for cadmium-treated roots was supplied with 150 µM CdCl_2_. Excised root tips were fixed in PBS-buffered 4% paraformaldehyde (4 °C; pH 7.4) for 40 min, and macerated for 15 min with citrate-buffered 2.5% pectinase (Sigma-Aldrich), at pH 5.0. Meristems squashed onto microscope slides (Polysine™, Menzel-Gläser, Germany) were air dried and, after washing with PBS, nuclear DNA replication was visualized using the Click-iT DNA Alexa Fluor^®^ 555 Imaging Kit (Thermo Fisher Scientific), according to the supplier’s instructions. Slides washed with PBS were stained for 15 min using 15 μM 4′,6-diamidino-2-phenylindole (DAPI; Sigma-Aldrich). After washing with PBS, slides were mounted in PBS/glycerol/DABCO (2.3% diazabicyclo[2.2.2]octane) mixture.

### 2.5. Immunocytochemical Detection of Phosphorylated H3 (Ser10) Histones

Immunofluorescence-based detection of H3 histones phosphorylated at the conserved serine 10 residue was performed according to the methods described in [23]. Apical parts of *V. faba* roots (1.5-mm-long) excised from the untreated (control) and all cadmium-treated seedlings were fixed for 45 min (20 °C) in PBS-buffered 3.7% paraformaldehyde, washed with PBS, and placed for 45 min (37 °C) in a citric acid-buffered digestion solution (pH 5.0) containing 2.5% cellulase (Onozuka R-10; Serva, Heidelberg, Germany), 2.5% pectinase (Fluka, Germany), and 2.5% pectolyase (ICN, Costa Mesa, CA, USA). After washing with PBS and distilled water, root meristems were squashed onto slides and air dried. Slides were pretreated for 50 min with PBS-buffered 8% bovine serum albumin (BSA) and 0.5% Triton X-100 (20 °C) and incubated for 12 h in a humidified atmosphere (4 °C) with rabbit polyclonal anti-H3S10Ph antibody (1:500; Sigma-Aldrich, Poznan, Poland), dissolved in PBS containing 1% BSA, according to methods described earlier [23,25]. After washing with PBS, slides were incubated for 1.5 h (20 °C) with secondary goat anti-rabbit conjugated to Alexa Fluor^®^ 488 antibody (1:500; Cell Signaling) dissolved in PBS (1:500) and counterstained with propidium iodide (PI; 0.3 mg mL^−1^). Slides washed with PBS were air dried and embedded in a PBS/glycerol/DABCO mixture.

### 2.6. Immunocytochemical Detection of γ-Phosphorylated H2AX Histones

For all experimental series, detection of γ-phosphorylated H2AX histones was carried out using an antibody raised against a peptide with phospho-serine 139. Root meristems excised from *V. faba* seedlings were fixed for 45 min (20 °C) in PBS-buffered 3.7% paraformaldehyde and, after washing with PBS, incubated for 45 min (37 °C) with citric acid-buffered digestion solution (pH 5.0) containing 2.5% pectinase, 2.5% cellulase, and 2.5% pectolyase (Sigma-Aldrich). After washing with PBS and distilled water, root tips were squashed onto Super Frost Plus glass slides, air dried, and pretreated for 50 min (20 °C) with PBS-buffered 8% BSA and 4% Triton X-100 (Fluka). Incubation with a rabbit polyclonal antibody raised against human H2AX histones phosphorylated at Ser139 (at a dilution of 1:750), dissolved in PBS containing 1% BSA, was performed in a humidified atmosphere (4 °C) for 12 h. After washing with PBS, slides were incubated for 1.5 h (20 °C) with the secondary goat anti-rabbit conjugated to Alexa Fluor^®^ 555 antibody (1:500; Cell Signaling) in PBS (1:500) and counterstained with DAPI (0.4 mg mL^−1^). Slides washed with PBS were air dried and embedded in PBS/glycerol/DABCO mixture, as described earlier.

### 2.7. Immunocytochemical Staining of H4K5 Acetylated Histones

For immunocytochemical detection of H4 histones acetylated at lysine 5, *V. faba* roots (in all experimental series) were fixed for 45 min (20 °C) in PBS-buffered 3.7% paraformaldehyde. Excised root meristems were washed with PBS, placed in a pectinase/cellulase/pectolyase digestion solution, and squashed onto slides (according to methods described above) Following pretreatment with PBS-buffered 8% BSA and 0.1% Triton X-100 for 50 min (20 °C), slides were incubated with mouse polyclonal anti-H4K5Ac antibodies (Sigma-Aldrich), dissolved in PBS containing 1% BSA at a dilution of 1:500. After an overnight incubation (4 °C), slides were washed with PBS and incubated for 1.5 h (18 °C) with secondary goat anti-mouse IgG conjugated to Alexa Fluor^®^ 488 antibody in PBS (1:500; Cell Signaling). Cell nuclei were counterstained with propidium iodide and embedded (as before).

### 2.8. AgNOR Staining

AgNOR impregnation of the control RAM cells was performed using two modified procedures described by Gimenéz-Abián et al. [26] and by Howell and Black [27]. In the first method [26], *V. faba* seedlings were mounted for 8 h in 0.3% colchicine, washed with cold PBS, and squeezed between two microscope slides to isolate interphase cell nuclei and M-phase chromosomes. The obtained suspension was dropped onto Super-Frost microscope slides, dried, and fixed with freshly prepared 3:1 methanol–acetic acid (3:1, *v*/*v*). Slides were dried and immersed in 2× SSC at 60 °C for 45 min, rinsed in tap water and distilled water, and dried again. The freshly prepared 1 mL of the staining mixture consisting of 0.05% formic acid and 0.5 g of AgNO_3_ was dropped onto microscope slides (100 mL^−1^ per slide). After covering with a coverslip, slides were put in a wet chamber at 80 °C for 3 min, washed with distilled water, dried, and mounted (Canada balsam).

For the second procedure [27], RAMs were prefixed for 10 min with Sörensen’s phosphate-buffered (pH 7.2) 2% glutaraldehyde (Sigma-Aldrich) and then postfixed with ethanol–acetic acid mixture (3:1; *v*/*v*) for 5 min (4 °C). After fixing, root meristems were washed with ethanol, rehydrated, rinsed with distilled water, and mounted in a mixture consisting of 2% gelatine in 1% formic acid and 50% AgNO_3_ (1:2; *v*/*v*). The staining procedure was performed at 60 °C for 7 min. Stained RAMs were treated with 5% sodium thiosulfate (3 times), squashed on the microscope slides and, after removing the coverslips, washed with distilled water, dried, and embedded in Canada balsam.

### 2.9. Observations and Analyses

Observations were made using an E-600 epifluorescence microscope (Nikon) equipped with phase-contrast optics, U2 filter (UVB light; λ = 340–380 nm) for DAPI, B2 filter (blue light; λ = 465–496 nm) for Alexa Fluor^®^ 488, or G2 filter (green light; λ = 540/25 nm) for Alexa Fluor^®^ 555, and PI-stained cell nuclei. In each study, images were recorded at exactly the same time of integration using a DS-Fi1 CCD camera (Nikon). Quantitative analyses and nuclear DNA fluorescence measurements were made after converting color images into grayscale and expressed in arbitrary units as mean pixel value (pv) spanning the range from 0 (dark) to 255 (white) according to described methods [23,24]. The obtained data were expressed as the mean values ± standard deviation of the mean (±SD). Student’s *t*-tests for paired data were used to compare individual variables. All immunofluorescence analyses were repeated (at least twice), and the total number of analyzed cells for one set of data was always more than 1000.

## 3. Results

### 3.1. Cd(II) Ions Generate Hydrogen Peroxide and γ-Phosphorylation of H2AX Histones

Cadmium-induced stress in RAM cells of *V. faba* was found to be associated with an increased generation of hydrogen peroxide (H_2_O_2_), which was evidenced using diaminobenzidine (DAB) polymerization into reddish-brown deposits (Figure 1A–E). No or only slight DAB staining could be observed in root tip cells of the untreated seedlings (Figure 1A). As compared with the control, 24 h incubation with CdCl_2_ gave rise to the typical brow coloration of the cytoplasm and the nucleus (Figure 1B). Quite similar effects of H_2_O_2_-dependent oxidative DAB polymerization were observed in plants exposed to post-cadmium water incubations for 12 h (Cd-rec-12; Figure 1C) and 24 h (Cd-rec-24; Figure 1D,E).

Cellular factors engaged to counteract adverse effects of stress at the nuclear DNA level, such as surveillance and control mechanisms responsible for genomic integrity, depend largely on the sensor phosphoinositide 3-kinase (PIKK)-related kinases ATM and ATR, both capable of recognizing DNA damage or replication block [28,29], and on DNA-dependent protein kinase (DNA-PK; [30]), which is crucial for the repair of DNA double-strand breaks (DSBs). The role of ATM in molecular systems activated by genomic instability or DNA damage was proven by colocalization of ATM with the phosphorylated core H2A histone variant (γ-H2AX) in close proximity to DNA DSBs [31].

The relative number of cells showing nuclear foci of γ-H2AX histones (Figure 2) increased from less than 5% in the control to 91% in CdCl_2_-treated RAMs. In experimental series with post-cadmium water recovery incubations (Cd-rec-12 and Cd-rec-24), the labeling index dropped to 62% and 45%, respectively. Immunofluorescence observations indicated that the intranuclear distribution of γ-H2AX foci was largely the same, representing an evenly scattered pattern across chromatin in all phases of the cell cycle (Figure 2A–D). In CdCl_2_-treated RAMs (Figure 2B), the average number of distinct γ-H2AX foci in mid-S-phase cell nuclei (determined by microdensitometry after DAPI staining) was found to be about 14 times higher than that in the control samples (Figure 3). Both post-cadmium water recovery periods brought about some decrease in DSB-specific H2AX phosphorylation (Figure 2C,D) and reduced the mean number of immunofluorescence foci to less than 20 per single cell nucleus (Figure 3).

### 3.2. Cadmium-Stress Increased Condensation of Chromatin Is Correlated with Ectopic Histone H3S10 Phosphorylation

As compared with the control (Figure 4A), root apical meristems in plants incubated for 24 h with 150 μM CdCl_2_ (Figure 4B) and those exposed to post-cadmium water recovery treatments (Cd-rec-12; Figure 4C and Cd-rec-24; Figure 4D) revealed significant changes in the overall structural organization of both interphase cell nuclei (Figure 4A–G) and mitotic chromosomes (Figure 4E,F). Apart from the general increase in DNA condensation, Feulgen staining showed a spotted pattern of compact chromatin regions scattered within the nucleoplasm (Figure 4B–D,F), which was also clearly reflected by a comparison of microdensitometric plots (Figure 4G). Significant CdCl_2_-induced changes in chromatin structure were detected in mitotic figures (relatively rare), characterized by an abnormal arrangement and atypical condensation of the chromosomes (Figure 4F, see also Figure 4D, arrow).

Considering both an increased chromatin condensation and our earlier results demonstrating marked changes in histone H3S10 phosphorylation caused by DNA stress conditions [23,25], *V. faba* RAM cells from CdCl_2_-treated seedlings and those from plants exposed to post-cadmium water incubations (Cd-rec-12 and Cd-rec-24) were immunolabeled with anti-H3S10Ph antibody and Alexa Fluor (488)-conjugated secondary antibody (Figure 5). In contrast to the control cells (Figure 5A) showing histone H3S10 phosphorylation exclusively restricted to centromeres of mitotic chromosomes (exemplified by metaphase; Figure 5A Met), all root tip cells from plants exposed to CdCl_2_ treatments revealed specific immunofluorescence confined not only to successive periods of transition through M-phase (Figure 5B Met, 5C Pmet, and 5D Met) but also scattered randomly throughout the cell nucleus in all stages of interphase (Figure 5B–D, exemplified by G1 and G2 phases). Furthermore, M-phase cells from CdCl_2_-treated plants were characterized by more intense centromeric staining, and H3S10Ph signals were also localized to the more distant parts of the chromosomes (Figure 5B Met, Figure 5C Pmet, and Figure 5D Met). The experimental procedures employing post-cadmium water recovery incubations (Cd-rec-12 and Cd-rec-24; Figure 5C,D, respectively) revealed a decrease in the frequency and mean size of H3S10Ph immunolabeled chromatin areas and their preferential occurrence at perinucleolar regions (observed mainly after the first period of post-cadmium recovery in water; Figure 5C G1 and G2).

### 3.3. Cadmium-Induced Changes in DNA Replication and S-Phase Progression

Despite no obvious differences in relative shares of EdU-labeled S-phase nuclei selected in populations of *V. faba* root meristem cells analyzed in our study (Figure 6), a more detailed examination revealed an evident effect of CdCl_2_ on DNA replication. By combining microfluorimetric quantitation of DNA contents in DAPI-stained cell nuclei (shown in Figure 6A–D,A’–D’) and visual EdU fluorescence analysis discerning characteristic patterning of late-replicating heterochromatin, two parameters were measured to determine the major features of S-phase progression in CdCl_2_-treated root tips and those from both post-cadmium water recovery experiments.

First, as shown in Figure 7, proportions of early-, mid-, and late-S-phase cells, indicated roughly by weak (faintly speckled), strong (almost homogenous), and punctate fluorescence patterns of EdU labeling, underwent dynamic changes, depending on the applied treatment. The frequencies representing different subpopulations of S-phase cells in the control seedlings were found to be drastically modified in RAM cells replicating in the presence of CdCl_2_. Quantitative sampling of EdU-labeled RAM cells from cadmium-treated plants indicated a strong dominance of nuclei (about 88%) characterized by a low level of fluorescence, and only minor fractions of cells (about 4% each) characterized by an intense or heterochromatin-specific incorporation of EdU. Following post-cadmium water recovery treatments (Cd-rec-12 and Cd-rec-24), representation of such cells was found to be restored, which resulted in an increase in the relative number of strongly labeled cell nuclei in both experimental series (Figure 7).

The second parameter estimated by means of EdU incorporation was aimed at determining differences in the intensities of DNA replication (Figure 8). As compared with the selected mid-S-phase RAM cells in the control plants, the corresponding subpopulation of root meristem cells from the CdCl_2_-treated seedlings displayed a significantly lower level of fluorescence (about 2/3 of the control mean value). Contrarily, both post-cadmium water recovery periods (Cd-rec-12 and Cd-rec-24) were characterized by an evident increase in mean intensity of EdU incorporation. Although not quantified by microfluorimetric analyses, visual observations of S-phase cells representing early and late stages of DNA replication revealed (proportionally) similar changes in the intensities of DNA synthesis, parallel to those recorded for mid-S-phase cell nuclei tested in all experimental series.

### 3.4. Cadmium Stress Reduces Mitotic Activity of RAM Cells and Generates Chromosomal Defects

The 24 h incubation with CdCl_2_ brought about a significant decrease in the frequency of cell divisions, reflected in an almost 3 times lower mitotic index (MI; Figure 9). There was an even more pronounced drop in MI values estimated in seedlings exposed to post-cadmium water treatments (24 h CdCl_2_ + 12 or 24 h H_2_O recovery series, Cd-rec-12 and Cd-rec-24). In plants incubated solely with CdCl_2_ and those post-treated for 12 h with water, a large percentage of M-phase cells (27% and 33%, respectively) revealed subtle or more pronounced changes in chromosomal morphology, such as chromatid gaps, breaks, and entanglements (see also Figure 4F). Importantly, in seedlings incubated with CdCl_2_ and then shifted back to water for 24 h (Cd-rec-24), only a minor fraction of mitotic RAM cells revealed chromosome abnormalities, thus decreasing the aberration index (AI) to merely 2.5% (Figure 9).

### 3.5. Acetylation of H4K5

The data presented earlier by Jasencakova et al. [32,33] suggest that H4 acetylation of lysine 5 in nucleosomal histone H4 (H4K5Ac) is linked primarily with DNA replication. Our current results indicate that variable relationships between H4K5 acetylation (Figure 10 and Figure 11) and successive stages of the cell cycle (Figure 12) can be strongly modified by cadmium-induced stress conditions, which manifests in changing proportions of cell nuclei characterized by dominant immunofluorescence of either decondensed chromatin (euchromatic-type nucleus), intense labeling of condensed chromatin (heterochromatic-type nucleus), and/or nucleolar domains.

Although there are no clear-cut differences between the various experimental series with respect to the overall level of immunofluorescence evidenced by microscopic examination (Figure 10A–D), a more detailed insight into individual cell nuclei stained using anti-H4K5Ac antibody revealed a plethora of nuclear labeling patterns (Figure 11A–L). These include cell nuclei characterized by various levels of decondensed chromatin staining (euchromatic type; Figure 11A–D) and heterochromatin staining (Figure 11E), cell nuclei with unlabeled (Figure 11A,C–E) and intensely labeled nucleoli (Figure 11F–I), and intermediate types of nuclei. In most cases, strong fluorescence of nucleoli was linked with more or less intense staining of euchromatin (Figure 11F–I) and only rarely with heavily labeled heterochromatin domains (Figure 11I). In mitotic cells, strong immunofluorescence seems to be localized to the centromeric regions of chromosomes (Figure 11L).

Considering the number of visible fluorescence spots, two in prophase and metaphase (Figure 11J,K, respectively) and four in anaphase (Figure 11L), it is reasonable to suppose that, topologically, they correspond to brown chromosomal nucleolus-organizing regions (NORs) resolved using the silver staining procedure (AgNOR; Figure 11M,N). Small argyrophilic spots could also be discriminated within faintly colored mitotic cell nuclei observed in RAM cell populations processed by the conventional method for nucleolar staining (Figure 11O, arrows).

For quantitative purposes, three types of labeling were selected using H4K5Ac antibodies: (1) cell nuclei with more or less homogenous immunostaining, including those with faint dot-like textures typical of euchromatin (Figure 12A), (2) cells with large fluorescent heterochromatin areas, sometimes characterized by a typical Rabl configuration (Figure 12B), and (3) cell nuclei with a pronounced or predominant immunostaining of nucleoli, which was associated mainly with more or less strong labeling of euchromatin (Figure 12C). In effect, the criteria of nuclear classification adopted for quantitative analysis, based on the most typical categories of nuclear H4K5Ac immunostaining (depicting general tendencies), encompass the majority of but not all RAM cells.

The data presented in Figure 12A,B reveal a general effect of CdCl_2_ (including post-cadmium water incubations) consisting in a marked reduction in the frequencies of cell populations recognized as those characterized by a predominant immunostaining of both eu- and heterochromatic nuclear regions. As compared with the control population, this result indicates that there is a noticeable drop in the intensity of lysine 5 acetylation in H4 histones induced by cadmium (Cd(II)) ions. Clearly, however, early stages of the cell cycle (including mid-S-phase cells) in the cadmium-untreated root meristems are represented mostly by cells with nuclei with heavily stained decondensed (euchromatic type) areas of the nucleoplasm (Figure 12A). Contrarily, in all experimental series, the increased populations of cell nuclei with strong immunofluorescence of heterochromatin were found to be located mainly at later stages of interphase (Figure 12B).

Another regularity, which is opposite in character, can be seen when considering the relationship between the diagrams representative of cells with preferential immunostaining of heterochromatin (Figure 12B) and the diagrams showing cell cycle-dependent distribution of cell nuclei with intensely labeled nucleoli (Figure 12C). In the control, CdCl_2_-treated, and both post-cadmium water-incubated RAM cell populations, the lowest frequencies of cells with preferential nucleolar staining coincide with the cell cycle stages characterized by the most frequent representations of cells with heavily stained areas of heterochromatin. To some extent, the same inverse correlation can be seen with respect to cell nuclei with the euchromatic pattern of H4K5Ac immunostaining (Figure 12A).

## 4. Discussion

Reactive oxygen species (ROS) are unavoidably and continuously produced in plants as by-products of aerobic metabolism, including both respiration and photosynthesis [34]. Despite changing concepts regarding the roles and biological importance of active oxygen metabolites, having shifted to universal signaling rather than their damaging effects [35], the dominant view still remains the same and points to the dual, concentration-dependent character of ROS. Even at low doses, ROS are believed to be involved in signal transduction pathways (e.g., in response to pathogen attack); the overproduction of ROS and disruption of redox homeostasis brings about interactions with all kinds of organic molecules [36]. Cadmium (Cd(II) ions), among other heavy metals, has been considered a dangerous stress factor that contributes to the disturbance of the delicate balance between ROS production and scavenging. Despite a vast number of studies supporting the pathophysiologic relevance of cadmium (it has been listed as one of the most hazardous substances in the environment), the mechanism of Cd^2+^ toxicity in plants is not yet fully understood. Earlier studies have shown that cadmium activates NADPH oxidase localized in the membrane, which in turn contributes to an increase in the production of ROS and, hence, to the formation of oxidative stress [37,38]. In consequence, the resultant oxidative damage may result in the inhibition of growth and development, disturbances in physiological processes, and increased rates of DNA damage and cell death [39].

Cadmium binding to the sulfhydryl group inactivates thiol-containing enzymes [40] and blocks cellular mechanisms designed to remove H_2_O_2_, i.e., the glutathione/glutathione reductase (GSH/GR) system, catalases (CAT), and ascorbate peroxidases (APX), thus increasing the role of superoxide dismutases (SOD) responsible for H_2_O_2_ production [41]. Our histochemical analyses using DAB staining of *V. faba* RAM cells showed that 24 h treatment of young seedlings with CdCl_2_ significantly increased H_2_O_2_ production, which was also observable after 12 and 24 h recovery incubations in water. Similar results were obtained previously in studies on roots in alfalfa [42], *Solanum nigrum* L. [43], soybean [44], pea roots and leaves [45,46], BY-2 tobacco cells [47], rice seedlings [48], and other plant models. Although no definite answer concerning the exact functions of antioxidant enzymes in plants exposed to high and low concentrations of Cd^2+^ ions has yet been offered and there is still no universal model to define the cadmium–oxidative stress relationship, it seems possible that the high content of H_2_O_2_ observed in RAM cells exposed to Cd^2+^ ions and to post-cadmium incubations in water reflects the reduced activity of antioxidant enzymes.

At the cellular level, oxidative stress (OS) brings about a variety of DNA damage, including base and sugar changes, DNA–protein crosslinks, cyclization of sugars, and DNA interstrand crosslinks [49,50,51]. Structural modifications of DNA, in turn, alter its coding properties and affect processes such as DNA replication and transcription. Phosphorylation of histone H2AX on Ser-139 (γ-H2AX) is regarded as one of the earliest responses involved in cellular perception/signaling pathways activated by DNA double-strand breaks (DSBs) [29,52,53,54]. By using immunofluorescence labeling of γ-H2AX as a highly specific and sensitive molecular marker for monitoring OS-associated DNA damage [55], our current observations showed an increased number of visible γ-H2AX foci in RAM cells of *V. faba* exposed to CdCl_2_, corresponding well to the results obtained previously in studies on *V. faba* root meristems treated with inhibitors of DNA replication, aphidicolin and hydroxyurea [29], and in onion root meristem cells exposed to β-lapachone [56]. In contrast to persistent DAB staining, RAM cells in seedlings incubated with water (after both post-cadmium water recovery periods) revealed decreased levels of H2AX phosphorylation, manifested by reduced numbers of immunofluorescence foci per cell nucleus. A possible explanation for this may be attributed to the activation of cell cycle checkpoint factors, which concentrate in the aggregates of γ-phosphorylated H2AX histones and share overlapping locations with proteins critical for both recognition and repair of nuclear DNA damage sites (e.g., [31,57]). In effect, each incidence of structural damage or functional disorder at the genomic level may trigger checkpoint-dependent activities that block initiation or progression of DNA replication and prevent cells from entering mitosis before the completion of the S-phase. This type of inhibition may be of a temporary character (which is evidenced in our “water recovery” experiments) but, in some cases, checkpoint activation may lead to senescence or to cell death by apoptosis [58,59]. Furthermore, block or delay of the cell cycle progression may severely increase the time needed for the start of gene expression, indispensable to synthesize and to activate DNA damage repair factors [60,61].

In this context, it is also worth noting that in mammalian G2-phase cells (but not G1-phase cells), γH2AX-marked sites of DNA damage have been found to be colocalized with large H3S10 phosphorylated nuclear areas [62]. The spatial relationship between both histone modifications was reduced in γ-irradiation cells. Thus, although considering highly specific mechanisms of the cell cycle-regulated epigenetic changes of chromatin in plants (including phosphorylation target sites at histone H3), it cannot be ruled out that some γH2AX-H3S10 interaction takes place in *V. faba* RAM cells exposed to cadmium and to 12 and 24 h post-cadmium recovery water treatments, characterized by significant reductions in both phosphorylation levels.

A corresponding aspect brought up by Chakrabarti and Mukherjee [63] relates to the recently discovered cadmium-induced adaptive response (AR) function leading to the amelioration of the genotoxicity induced by ethyl methanesulfonate (EMS; a known chemical mutagen) in growing root cells of two model plant systems *Allium cepa* and *Vicia faba*. As compared with the untreated plants, root meristem cells of the Cd-conditioned plants were found to be protected against EMS challenge, which was evidenced by a number of cytogenotoxicity and oxidative stress parameters, including relative increase in cell viability, mitochondrial membrane potential mitotic index, relative decrease in DNA damage, micronuclei frequency, and chromosome aberrations. Additionally, another epigenetic modification, i.e., DNA methylation, has been proven to be essential for the activation of AR functions, thus implicating that a vast amount of molecular and physiologic processes must be integrated to orchestrate appropriate cellular functions capable of facing environmental stress conditions [63].

Our experiments with CdCl_2_-treated RAM cells of *V. faba* clearly demonstrate checkpoint control functions over both S- and M-phases of the cell cycle. Although incorporation of EdU, employed as a replicational marker, showed that cadmium ions did not change the relative numbers of cells active in DNA synthesis, at the same time, however, it revealed a strong reduction in the intensity of nuclear labeling, suggesting a decrease in the rate of DNA replication. This result corresponds well with other studies indicating that ROS can disrupt S-phase progression through different mechanisms, including replication fork (RF) arrest caused by oxidation of replication proteins and ROS-induced DNA damage [7], or by a decrease in the rate of RF movement per unit of time [64]. In fact, a number of research studies have demonstrated that a single-stranded DNA template at the RF is susceptible to oxidative base damage and DNA breaks [65], and that the oxidized bases can inhibit the progression of polymerases [66], cause RF collapse [7], and affect the initiation of DNA synthesis [67].

The results presented in this work seem to indicate that another cell cycle checkpoint is activated in response to either cadmium-induced oxidative stress or ROS-generated DNA DSBs. Incubation of *V. faba* RAM cells with Cd^2+^ ions (and in all experimental series with post-cadmium water recovery treatments) brings about a marked reduction in the mitotic activity. This effect may be caused either by (1) direct influence of cadmium on a large number of enzymatic and structural proteins required for an undisturbed process of chromosome condensation, (2) by the repression of genes encoding all indispensable elements needed to pass the G2-to-M-phase transition and to condense chromosomes, or (3) by the coherent system of inter-related cell cycle checkpoint functions in the S-phase and in the G2-phase (the latter being more prominent in plants) responsible for maintaining genomic integrity and monitoring the metabolic state and physiological condition of the cell prior to its transition into mitosis.

The decreased mitotic activity observed in cadmium-treated root meristems of *V. faba* is correlated with a specific type of chromatin restructuring that establishes a speckled character of Feulgen DNA-stained cell nuclei in interphase. Although not proven in evidence as causal, it seems possible that at least part of this coincidence may be due to Aurora B kinase-mediated phosphorylation of serine 10 on histone H3 (H3S10ph). In general, this modification is observed in two different epigenetic contexts, i.e., prior to mitotic and meiotic (II) chromosome condensation [68], and in interphase, during transient marking of inducible genes [69,70]. The latter process was found to be associated with the stress-stimulated activation of the MAPK pathway, followed by H3S10 phosphorylation at promoter and enhancer regions of immediate early genes and release of paused RNA POL II responsible for transcription elongation [71]. A broader, mitosis-independent role of histone H3S10 phosphorylation in the regulation of transcription and other chromatin-state dependent functions in interphase was postulated by [70] after revealing a significant fraction of the genome enriched with H3S10ph in both G1- and S-phases of the cell cycle. It should be emphasized, however, that in plants, mitotic H3S10 phosphorylation starts no earlier than in prophase (later than in vertebrates) and does not extend beyond the pericentric region of the chromosome [23,68].

Irrespective of the exact mechanism that leads to or originates from structural reorganization of chromatin in *V. faba* root meristem cells exposed to cadmium, all changes in their metabolic state are inevitably correlated with profound modifications at the proteomic and epigenetic levels. As shown in our studies, apart from phosphorylation of histone H3 (serine 10) and γ-phosphorylation of histone H2AX (serine 139), Cd^2+^ ions induced considerable changes in cell cycle-related acetylation of H4 histone at the N-terminal K5 residue. Earlier studies [72,73] concentrated mainly on rDNA genes in *V. faba*, and suggested that H4 acetylation is loosely associated with their transcriptional activity and inversely correlated with nucleolar rDNA replication during the early stages of the S-phase. Extensive changes in H4 acetylation, yet unrelated to cell cycle stages or to specific chromatin domains, were also observed in *V. faba* and barley (*Hordeum vulgare*) in [74].

Our present experiments indicate that the distribution of histone H4 acetylated on lysine 5 in the untreated RAM cell nuclei of *V. faba* largely depends on both cell cycle position and intranuclear localization within the territories occupied by either euchromatin, heterochromatin, or the nucleoli. Furthermore, a temporal spread of epigenetic markers throughout interphase (from early G1- to late G2-phase) allows us to suggest that acetylation of H4K5 may have an essential regulatory role in DNA replication as well as in transcription. Each of these processes requires a transient unwinding of the DNA duplex and separation of its two strands, making them extremely susceptible to ROS-mediated damage. This, in fact, may be the cause of the cadmium-induced erosion of epigenetic markers observed in the S-phase (during nuclear DNA synthesis) and in the G1- and G2-phases of the cell cycle (during post-M- and post-S-phase transcriptions). Apparently, the process of nucleolar H4K5 acetylation in RAM cell nuclei of *V. faba* is considerably less vulnerable to cadmium-induced stress than eu- and heterochromatin areas of the nucleoplasm. According to [32] (interpretation based on the results obtained by other authors), such phenomena may be explained, in part at least, by the fact that nucleolar territories are deprived of nucleosomes. On the other hand, epigenetic markers established by H4K5Ac may be viewed as a kind of “cell memory”, by which histones localized in the pericentromeric regions of mitotic chromosomes may transmit acetyl tags as signals for proteins required during the next interphase (in the G1-phase of the new cell cycle) for the start of early transcription and for metabolic reactivation [75]. Hypothetically, the same explanation might be applied for cells exposed to cadmium-induced stress. In this case, however, persistent acetyl marks observed within nucleolar territories would play a protective role in cell nuclei characterized by the ectopic appearance of clumps of compact chromatin, considerably less open and less favorable for transcription. Accordingly, our preliminary studies using 5-ethynyl-2′-deoxyuridine incorporation followed by Click-iT Alexa Fluor 488 imaging revealed a considerable decrease in the quantities of nascent RNA produced in the cadmium-treated RAM cells of *V. faba*, compared with the control plants (data not shown).

Considering all the effects reported above, RAM cells of *V. faba* exposed to the toxic effects of cadmium seem to respond by triggering a long series of interlinked biochemical reactions, which by ROS-mediated oxidative stress and DNA damage-induced replication stress ultimately activate signal factors engaged in cell cycle control pathways, DNA repair systems, and epigenetic modifications. Crosstalks between these mechanisms allow the creation of a coordinated and consistent strategy to neutralize the harmful effects of the stress conditions encountered by plant organisms in the natural environment.

## Figures and Tables

**Figure 1 cells-10-00640-f001:**
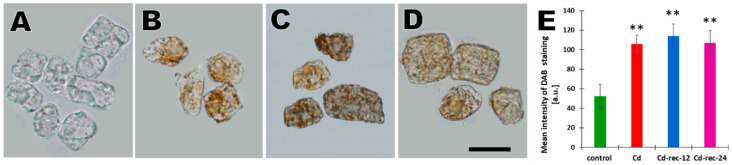
Generation of H_2_O_2_ visualized as brown deposits of oxidized diaminobenzidine (DAB) in the control seedlings (**A**), CdCl_2_-treated seedlings (**B**), and in plants exposed to 12 h (Cd-rec-12; (**C**)) and 24 h (Cd-rec-24; (**D**)) post-cadmium recovery incubations in water. Scale bar = 10 μm. (**E**) Cytometric evaluation of mean DAB staining intensities ([a.u.]—arbitrary units ± SD); ** indicates statistical significance at *p* < 0.002, as compared with the control.

**Figure 2 cells-10-00640-f002:**
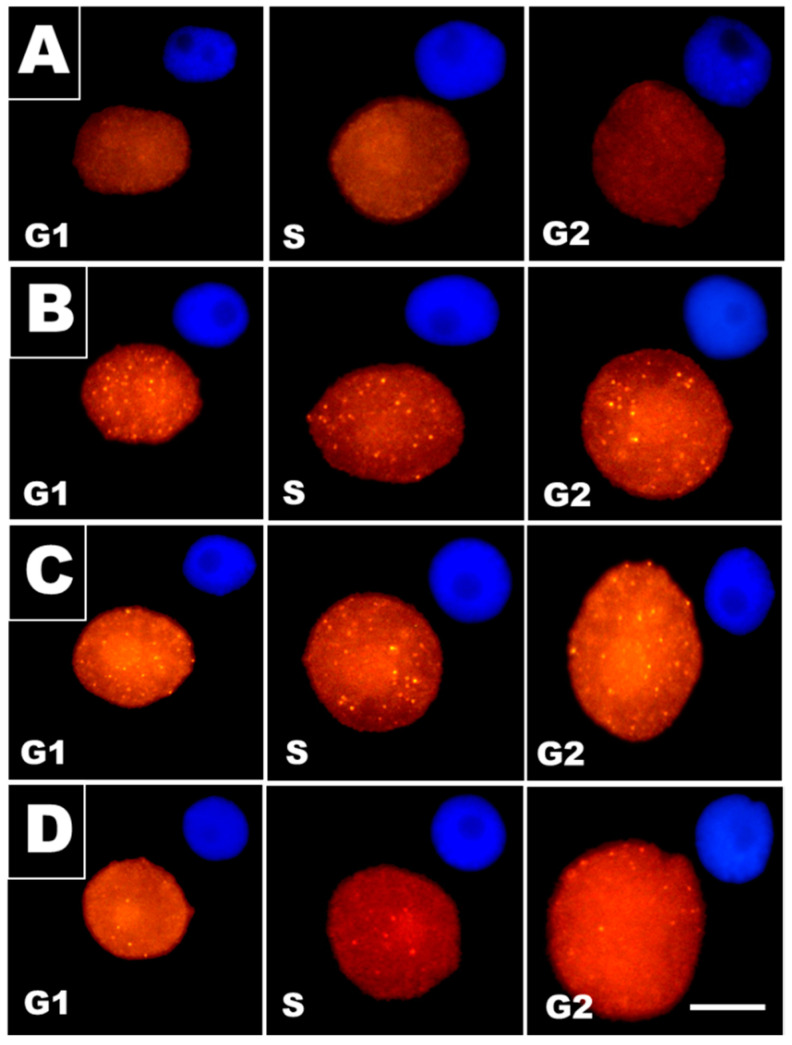
Immunofluorescent foci of γ-phosphorylated H2AX histones in G1-, S-, and G2-phase cell nuclei from root apical meristems (RAMs) of *V. faba* seedlings: (**A**) control, (**B**) CdCl_2_ treatment, (**C**) post-cadmium recovery with water for 12 h (Cd-rec-12), and (**D**) post-cadmium recovery with water for 24 h (Cd-rec-24). The smaller micrographs (size reduced by 50%) embedded in the upper right corners show corresponding images of 4′,6-diamidino-2-phenylindole (DAPI)-stained cell nuclei. Scale bars for micrographs showing γ-H2AX immunostaining = 10 μm.

**Figure 3 cells-10-00640-f003:**
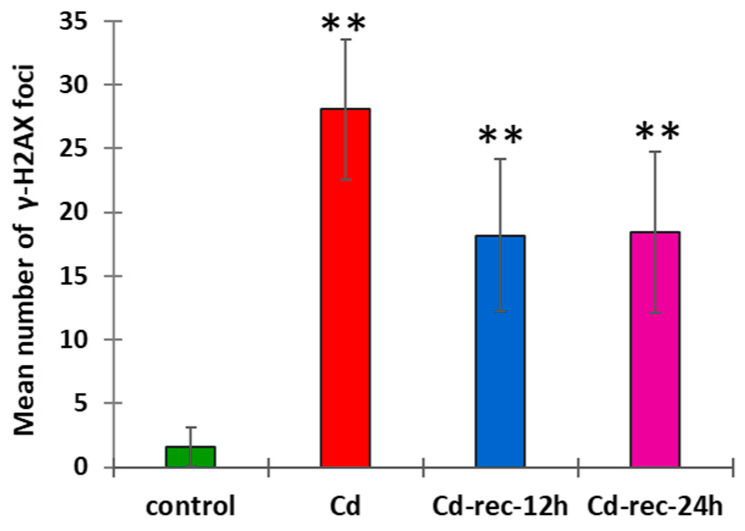
Mean number of intranuclear phospho-H2AX foci generated in *V. faba* mid-S-phase RAM cell nuclei from the control seedlings, CdCl_2_-treated plants (Cd), and plants exposed to post-cadmium recovery incubations with water for 12 h (Cd-rec-12) and 24 h (Cd-rec-24). As compared with the control, statistically significant changes in mean number of γ-H2AX foci (±SD) are marked by asterisks: ** indicates *p* < 0.005.

**Figure 4 cells-10-00640-f004:**
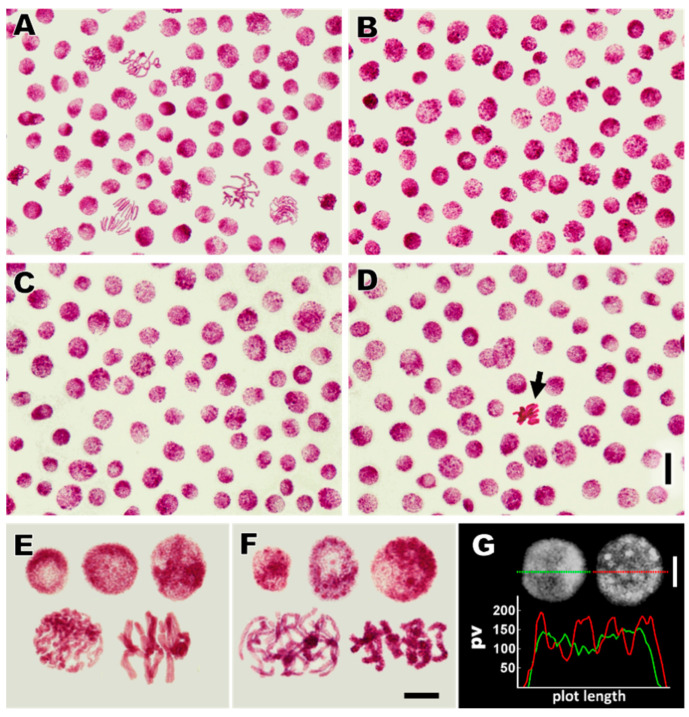
Feulgen DNA staining of *V. faba* RAM cells. (**A**–**D**) An overall view of cell population in root meristems from the control (**A**), CdCl_2_-treated seedlings (**B**), and post-cadmium recoveries with water for 12 h (Cd-rec-12; (**C**)) and 24 h (Cd-rec-24; **D**); scale bar = 20 μm. (**E**,**F**) Comparison of the interphase G1-, S-, G2-phase cell nuclei (upper rows from left to right, respectively) and mitotic cells during prophase and metaphase (lower rows from left to right) in the control (**E**) and cadmium-treated RAMs ((**F**), see also (**D**); arrow); scale bar = 10 μm. (**G**) Densitometric plots showing relative changes in chromatin condensation (expressed as pixel values (pv) from 1 to 255) in the negatives of cells from the control (left nucleus; green line) and CdCl_2_-treated (right nucleus; red line) root meristems. Scale bar = 10 μm; plot lengths are 20 μm for the control cell nucleus and 21.5 μm for the Cd-treated cell nucleus.

**Figure 5 cells-10-00640-f005:**
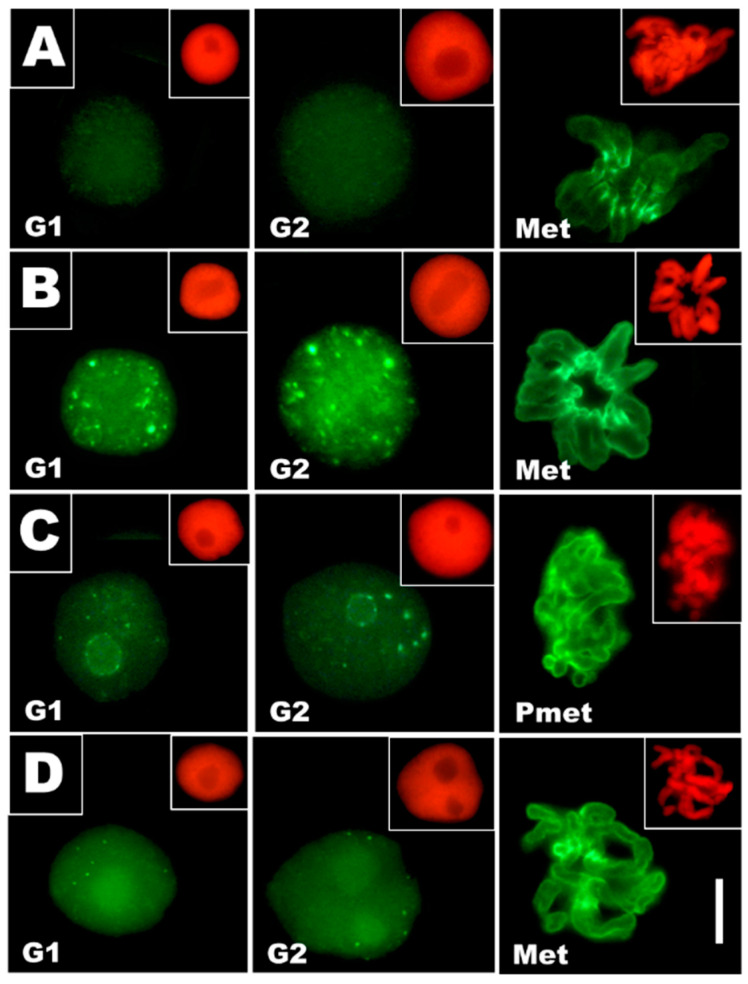
Immunofluorescence detection of phosphorylated histone H3S10 in *V. faba* RAM cell nuclei during early G1 and late G2 phases of interphase and during metaphase (Met) and prometaphase (Pmet), from the control seedlings (**A**), CdCl_2_-treated plants (**B**), and from plants exposed to post-cadmium recovery incubations in water for 12 h (Cd-rec-12; (**C**)) and 24 h (Cd-rec-24; (**D**)). Pericentromeric staining of chromosomes can be seen in mitotic cell nuclei. Scale bar = 10 μm. The smaller micrographs (size reduced by 50%) embedded in the upper right corners show corresponding images of cell nuclei stained with propidium iodide.

**Figure 6 cells-10-00640-f006:**
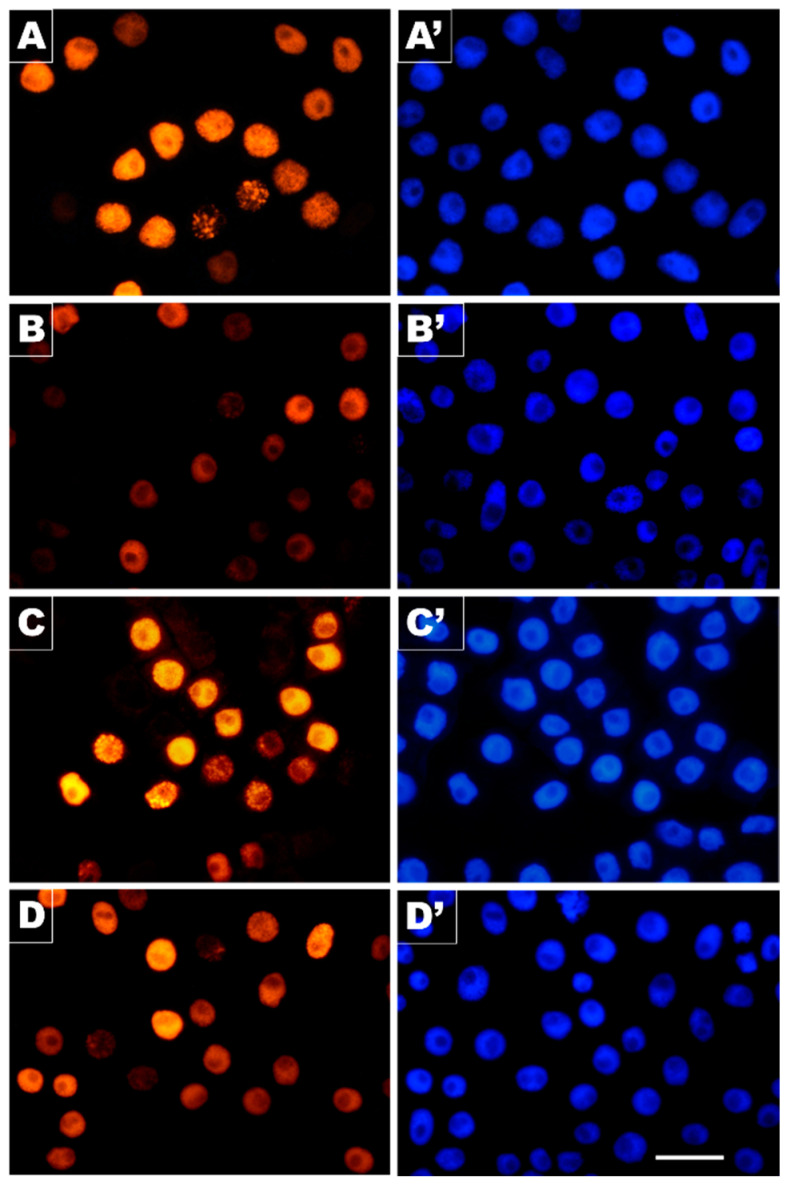
DNA replication in *V. faba* RAM cell populations evidenced by incorporation of EdU and copper(l)-catalyzed Alexa Fluor^®^ 555 azide-alkyne “click” reaction. (**A**) Control, (**B**) CdCl_2_ treatment, (**C**) exposition to post-cadmium recovery incubation in water for 12 h (Cd-rec-12), and (**D**) for 24 h (Cd-rec-24); (**A’**–**D’**): corresponding images of the same cell populations stained with DAPI. Scale bar = 50 μm.

**Figure 7 cells-10-00640-f007:**
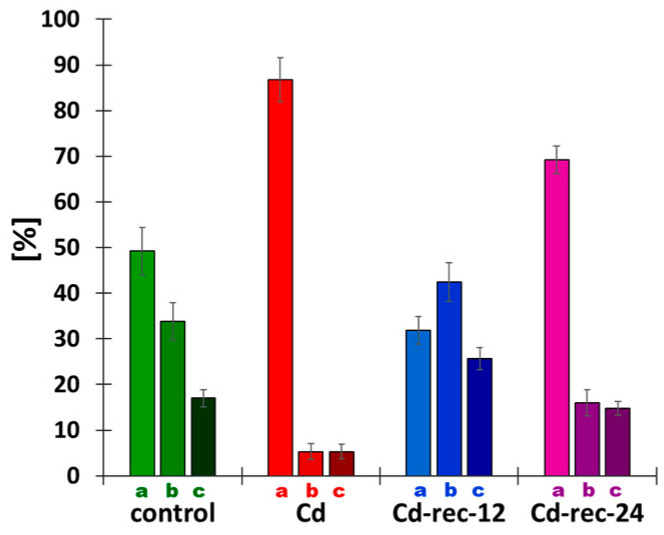
Fractions ([%] ± SD) of early-(a), mid-(b), and late-S-phase (c) cells in RAMs of *V. faba* seedlings, calculated by combining microfluorimetric quantitation of DNA contents in DAPI-stained cell nuclei and visual EdU fluorescence analysis. Diagram sets represent control seedlings, CdCl_2_-treated plants (Cd), and plants exposed to post-cadmium recovery incubations in water for 12 h (Cd-rec-12) and 24 h (Cd-rec-24).

**Figure 8 cells-10-00640-f008:**
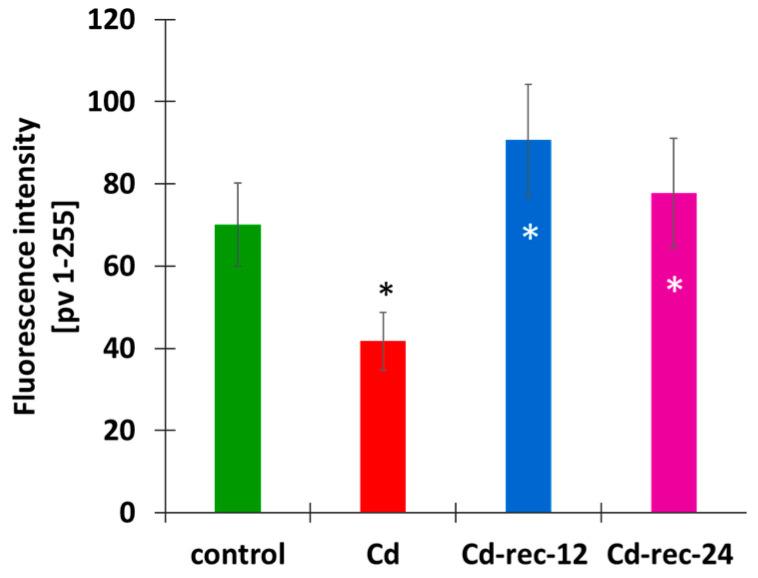
Mean intensity of EdU labeling in *V. faba* mid-S-phase RAM cell nuclei from the control seedlings, CdCl_2_-treated plants (Cd), and plants exposed to post-cadmium recovery incubations with water for 12 h (Cd-rec-12) and 24 h (Cd-rec-24). As compared with the control, statistically significant changes in the intensity of EdU labeling/fluorescence (±SD) are indicated by a black asterisk (*p* < 0.05); statistically significant difference was also noted in comparisons between cadmium-treated RAM cells and post-cadmium water recovery incubated plants (white asterisk, *p* < 0.01).

**Figure 9 cells-10-00640-f009:**
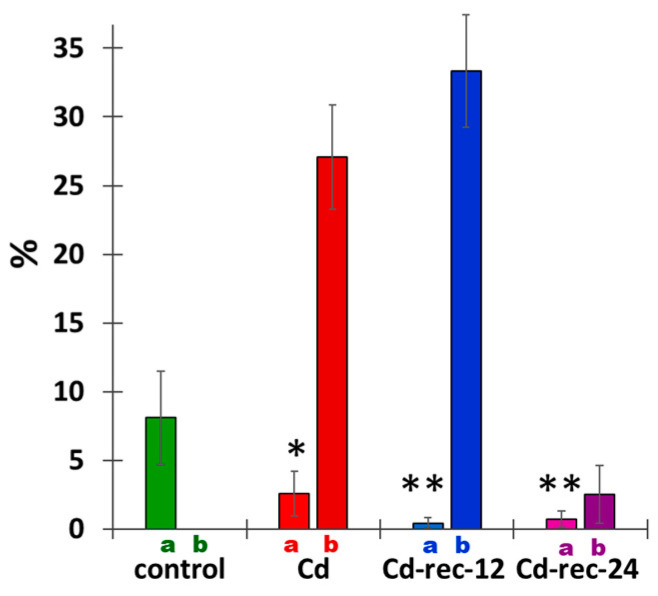
Mitotic index (a; %) and the fractions of aberrant M-phase cells (b; %) calculated for RAM cells in the control, CdCl_2_-treated seedlings (Cd), and in plants exposed to post-cadmium recovery incubations in water for 12 h (Cd-rec-12) and 24 h (Cd-rec-24). Mean values were calculated for ten root meristems (1000 cells per meristem). Compared with the control, statistically significant changes in mean mitotic index values (± SD) are marked by asterisks: * indicates *p* < 0.05, and ** indicates *p* < 0.005.

**Figure 10 cells-10-00640-f010:**
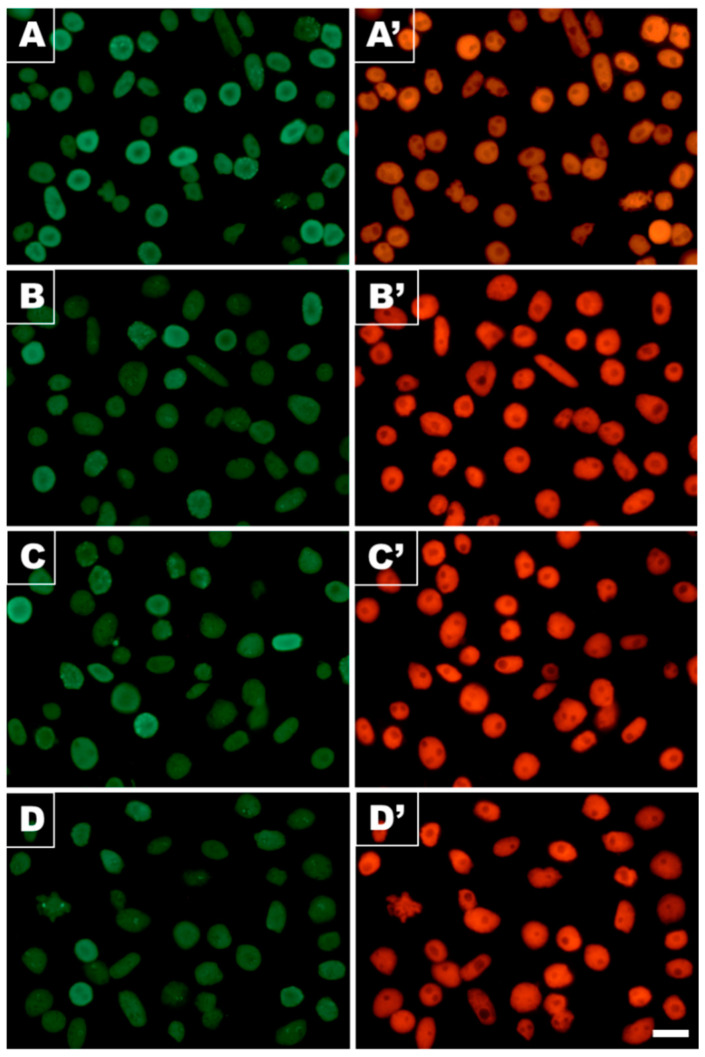
Immunofluorescence detection of acetylated histone H4K5 in *V. faba* RAM cell nuclei from the control seedlings (**A**), CdCl2-treated plants (**B**), and from plants exposed to post-cadmium recovery incubations in water for 12 h (Cd-rec-12; (**C**)) and 24 h (Cd-rec-24; (**D**)); (**A’**–**D’**): corresponding images of the same cell populations stained with propidium iodide. Bar = 20 μm.

**Figure 11 cells-10-00640-f011:**
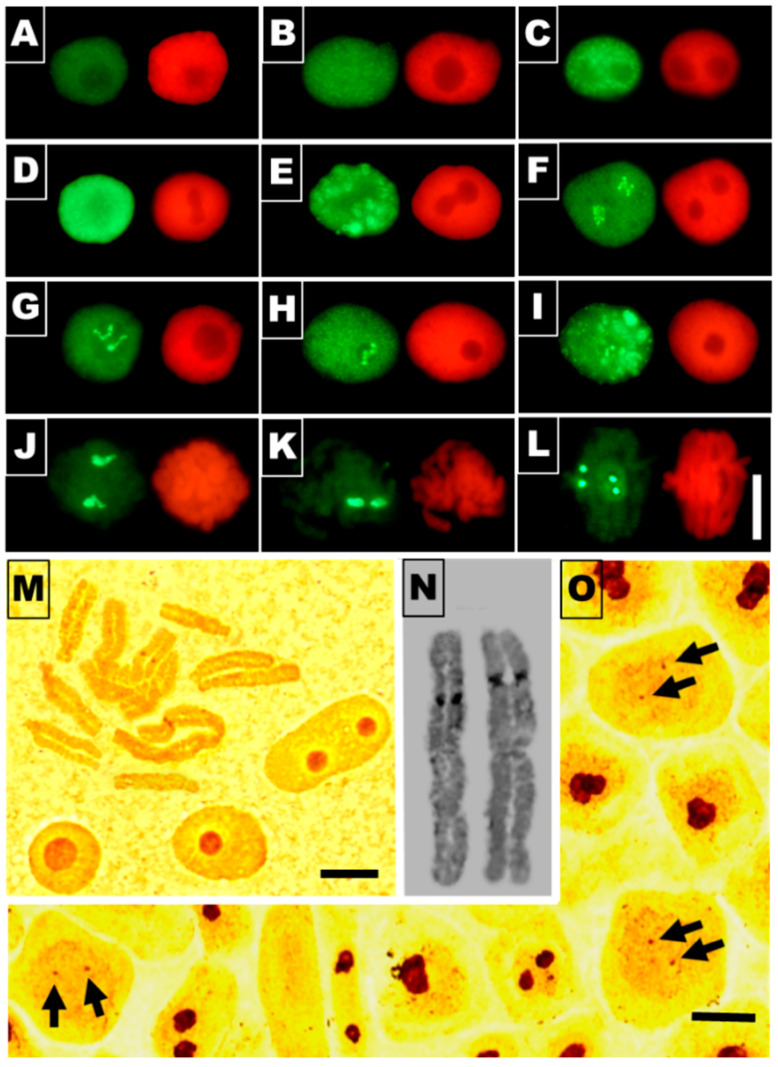
Different labeling patterns observed in RAM cell nuclei selected from all experimental series (control, CdCl2-treated, and post-cadmium water recovery incubations) after immunostaining with antibodies against K5 acetylated H4 histones (left side of each panel) and corresponding images of cell nuclei stained with propidium iodide (right side of each panel): (**A**–**D**) cell nuclei with increasing immunofluorescence labeling evenly spread over the chromatin; (**E**) cell nuclei with immunostained heterochromatin areas; (**F**–**H**) predominant nucleolar staining in cell nuclei of euchromatic type; (**I**) strong nucleolar staining in cell nucleus showing heterochromatic type of immunofluorescence; (**J**–**L**) immunofluorescence labeling of the chromosomal areas corresponding to the nucleolus-organizing regions (NORs) during prophase (**J**), metaphase (**K**), and anaphase (**L**). Bar = 10 μm. (**M**–**O**) AgNOR staining of the control *V. faba* RAMs: (**M**) isolated interphase cell nuclei and metaphase chromosomes stained using AgNOR procedure according to [26]; bar scale = 10 μm; (**N**) NOR-carrying chromosomes selected from two different metaphase plates; bar scale = 5 μm; (**O**) silver-stained RAM cell population stained using procedure in [27]; note brown spots (indicated by arrows) that may represent NORs in faintly stained mitotic cell nuclei; scale bar = 10 μm.

**Figure 12 cells-10-00640-f012:**
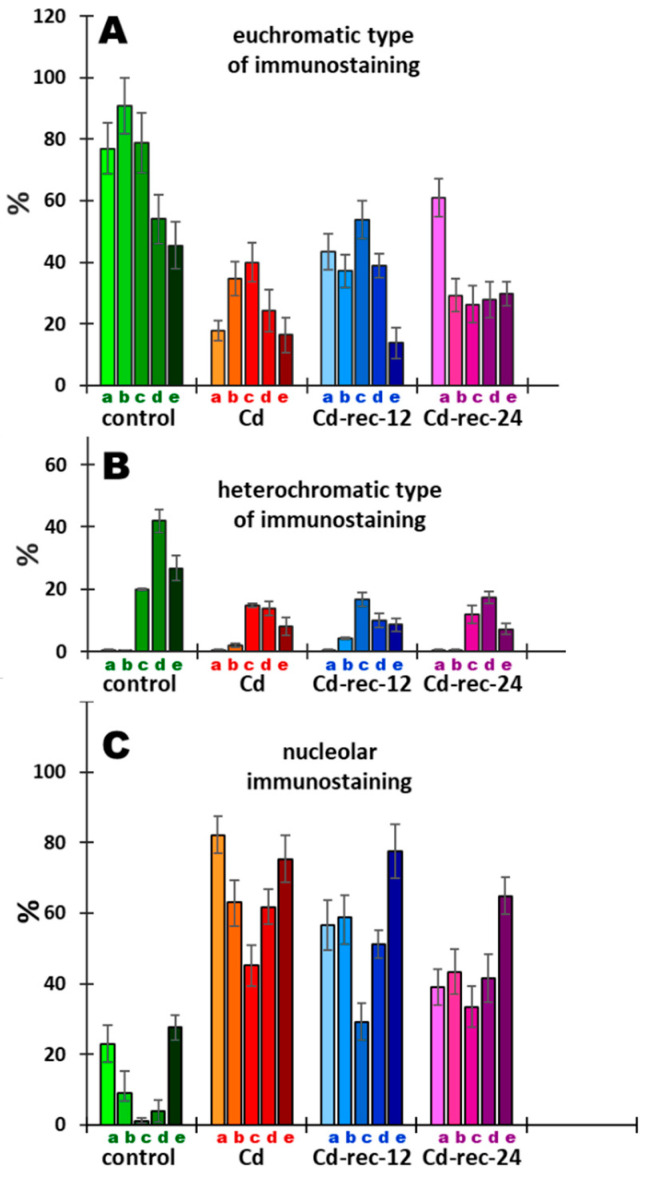
Frequencies (% ± SD) of nuclei with H4K5Ac immunolabeling localized preferentially in: (**A**) euchromatic regions, (**B**) heterochromatic regions, and (**C**) nucleolar regions, discriminated with respect to successive stages of the cell cycle in *V. faba* root meristems; control, CdCl_2_-treated seedlings (Cd), and plants exposed to post-cadmium recovery incubations with water for 12 h (Cd-rec-12) and 24 h (Cd-rec-24). Successive bars in the diagrams represent: **a**—G1-phase, **b**—early S-phase, **c**—mid-S-phase, **d**—late S-phase, and **e**—G2-phase. Categories of cells were selected according to immunofluorescence staining patterns presented in Figure 11A–L. Cells displaying near-background, extremely faint, weak, or undefined immunofluorescence were omitted from categorization.

## Data Availability

The data presented in this study are openly available.

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
