# Peer review of "Cadmium (II)-Induced Oxidative Stress Results in Replication Stress and Epigenetic Modifications in Root Meristem Cell Nuclei of Vicia faba"

_cells, 2021, doi:10.3390/cells10030640_

Round 1
Reviewer 1 Report
The MS presents interesting data and good figures, but it is not well written (especially the figure captions) and, apparently, the data for some theses are not shown. Therefore, it needs a thorough revision.
Specific points:
Introduction: Cd is not a microelement for plant; therefore, It should be made clear at the beginning of the introduction that some heavy metals are essential plant microelements that become toxic at high concentrations, whereas Cadmium is not among them and is considered a toxic element.
Lines 90-91: some words should be added regarding the Cd concentration applied because different Authors employ various Cd concentrations in their tests without say why.
The main critical point concern Figure 1 and following Figures including photos (Fig. 2, 4, 5, 6, 10): if there are 4 samples (Fig. 1 B, Control, Cd treated seedlings, Cd rec 12, Cd rec 24) the photos must also be 4. In fact, only the histogram figures include the four samples, whereas only Fig. 5 shows immunofluorescence of Cd rec 12, but with only two photos (and only one for Cd rec 24).
In addition, if the sub-figures are indicated with letters of the alphabet, within A (or B) further partitions should be distinguished with Arabic numerals (1, 2, 3 ...) or eventually with G1, S, G2, to allow for an immediate and simple understanding of the correspondence with what is written in the figure captions. Otherwise it is really difficult to understand the results
Figure 3: It is appropriate to colour the histograms by giving a different colour to each treatment and subsequently using gradients of the same colours for different development stages (Fig. 7 and 11) or different indices (Fig. 9).
The really recent article of Chakrabarti and Mukherjeen about the effect of Cd on Allium cepa and Vicia faba (Investigating the underlying mechanism of cadmium-induced plant adaptive response to genotoxic stress Ecotoxicology and Environmental Safety 2021, 209, 111817 https://doi.org/10.1016/j.ecoenv.2020.111817) must be cited and used in Discussion.
Author Response
March 8, 2021
Dr. Aneta Żabka*, Dr. Konrad Winnicki, Dr. Justyna Teresa Polit, Mateusz Wróblewski, Prof. Dr. Janusz Maszewski
University of Łódź
Faculty of Biology and Environmental Protection
Department of Cytophysiology
Pomorska 141/143
90-236 Łódź
Poland
*Corresponding author:
tel.: +48 42 635 45 12
e-mail: aneta.zabka@biol.uni.lodz.pl
Dear Reviewer 1.
We wish to express our gratitude for your comments and remarks. We have done our best to follow the views and advice received in your review. The following changes were made:
- Information concerning Cd was extended and we have added new citation in the text [3].
- The information concerning concentration of CdCl2 was added and two new citations were added to the text [20, 21].
- All tables included into our manuscript have been changed, in agreement with you suggestions, i.e., the number of photos was increased to 4 (Figures 2, 4, 5, 6, 10).
- Lettering merged in the figures was changed and considerably simplified, according to your suggestion.
- All histograms have been modified using different colour to each treatment (also using gradients of the same colours for different stages or indices, according to your suggestions).
- An inspiring article by Chakrabarti and Mukherjee [63]
Additionally, we have made several corrections in the text and introduced AgNOR staining, according to suggestions of Reviewer 2.
Yours faithfully,
Aneta Żabka and co-workers
Reviewer 2 Report
Authors studied epigenetics adaptation of DNA repair systems. Among others, they studied profiles of gammaH2AX, H3S10ph, and /or H4K5Ac under oxidative stress induced by heavy metal, Cadmium. As the experimental model, they used the plant model of Vicia faba. From this view, it is important to mention that Cadmium enters plants as a bivalent cation (Cd2+) absorbed by the roots through low-specificity Ca2+ transporters or channels. The description of this mechanism is very summarizing in this paper.
Here, I would like to suggest some points to be improved:
In Fig. 2, the authors show gH2AX distribution and levels in the cell cycle phases, but I would like to recommend showing an additional marker of G1/S and G2 phases. At least it should be H3S10ph having a distinct H3S10ph profile in G1/S and G2 phases. G1 from S is possible to recognize according to the morphology of PCNA (for example). Maybe, it is complicated to visualize PCNA in plants, but certain information was published, e.g., https://onlinelibrary.wiley.com/doi/full/10.1046/j.1365-313x.2001.01184.x
In Fig. 4c, the description of axes is incomplete.
Fig. 5, in mammalian cells, H3S10ph has its specific distribution profiles in distinct cell cycle phases (see for example https://www.sciencedirect.com/science/article/pii/ S0300908418302256), so one would expect that the same can be seen in the plant genome. The authors should discuss it, at least.
Fig. 10 NORs should also be visualized by Ag-staining to be sure of the morphology of NORs - AgNORs.
Fig. 11 In order to discriminate between euchromatin and heterochromatin, a better marker of heterochromatin should be used. It could be H3K9me3, and euchromatin can be visualized by anti-H4 acetylation or anti-H3K9ac - dual-labeling, for example.
A marker of DNA damage is not only gH2AX and H3S10ph, but the authors should also study H4K20me2/me3 as markers of non-homologous end joining (NHEJ).
Author Response
March 8, 2021
Dr. Aneta Żabka*, Dr. Konrad Winnicki, Dr. Justyna Teresa Polit, Mateusz Wróblewski, Prof. Dr. Janusz Maszewski
University of Łódź
Faculty of Biology and Environmental Protection
Department of Cytophysiology
Pomorska 141/143
90-236 Łódź
Poland
*Corresponding author:
tel.: +48 42 635 45 12
e-mail: aneta.zabka@biol.uni.lodz.pl
Dear Reviewer 2.
We wish to express our gratitude for your comments and remarks. Although some of the indicated points appeared unsupportable for us by experimental evidence, we have done our best to follow the views and advice received in your review. The following changes were made:
- According to our best knowledge, phosphorylation of H3S10 in plants is restricted to prophase and can not be used as a marker for the cell cycle phase. Our experience indicates, however, that H3S10Ph appears earlier under stress conditions (e.g. in G2 phase in cells exposed to DNA stress conditions induced by hydroxyurea or 5-aminouracil). However, considering your remarks, the specificity of H3S10 phosphorylation was emphasized in a new version of the text.
- Our affords towards discrimination between eu- and heterochromatin using H3K9me3 brought no satisfactory results. This may have resulted from faba specificity of RAM cells or other reasons. Unfortunately, it was not possible now to purchase suggested antibodies (anti-H4 acetylation, anti-H3K9ac and H4K20me2/3). Considering your comment regarding eu- and heterochromatin, however, we have modified the terminology, using “euchromatic-type”, “decondensed chromatin”, etc. Certainly, morphology is not a good marker for eu- and heterochromatin (genetic terms).
- The description of the axis in Figure 4 is changed.
- We have performed AgNOR staining (two methods) to strengthen our supposition concerning fluorescing points within mitotic chromosomes (K5 acetylated H4 histones).
Additionally, we have made several corrections in the text. 7 new citations were added, a new table was prepared.
Yours faithfully,
Aneta Żabka and co-workers
Round 2
Reviewer 1 Report
The MS was adeguately improuved
Reviewer 2 Report
I agree with the publication of this paper.